# AIDS Drug Assistance Program disenrollment is associated with loss of viral suppression beyond differences in homelessness, mental health, and substance use disorders: An evaluation in Washington state 2017–2019

Steven J. Erly[1,2]*, Christine M. Khosropour[1], Anjum Hajat[1], Monisha Sharma[3], Jennifer R. Reuer[2], Julia C. Dombrowski[1,4]

1 Department of Epidemiology, University of Washington, Seattle, Washington, United States of America, 2 Washington State Department of Health, Olympia, Washington, United States of America, 3 Department of Global Health, University of Washington, Seattle, Washington, United States of America, 4 Department of Medicine, Division of Allergy and Infectious Diseases, University of Washington, Seattle, Washington, United States of America

* steven.erly@doh.wa.gov

**Data Availability Statement:** The data used in this study is derived from state HIV surveillance data

## Abstract

AIDS Drug Assistance Programs (ADAPs) are state-administered programs that pay for medical care for people living with HIV in the US. Maintaining enrollment in the programs is challenging, and a large proportion of clients in Washington state (WA) fail to recertify and are disenrolled. In this study we sought to quantify the impact of disenrollment from ADAPs on viral suppression. We conducted a retrospective cohort study of the 5238 clients in WA ADAP from 2017 to 2019 and estimated the risk difference (RD) of viral suppression before and after disenrollment. We performed a quantitative bias analysis (QBA) to assess the effect of unmeasured confounders, as the factors that contribute to disenrollment and medication discontinuation may overlap. Of the 1336 ADAP clients who disenrolled ≥1 time, 83% were virally suppressed before disenrollment versus 69% after (RD 12%, 95%CI 9–15%). The RD was highest among clients with dual Medicaid-Medicare insurance (RD 22%, 95%CI 9–35%) and lowest among privately insured individuals (RD 8%, 95%CI 5–12%). The results of the QBA suggest that unmeasured confounders do not negate the overall RD. The ADAP recertification procedures negatively impact the care of clients who struggle to stay in the program; alternative procedures may reduce this impact.

## Introduction

The Ryan White HIV/AIDS Drug Assistance programs (ADAPs) are the largest source of medical care for people living with HIV (PLWH) in the United States and a critical part of the Treat Pillar in federal plans to reduce HIV incidence by 90% by 2030 [1]. ADAPs pay for insurance and medical care for 20% of people living with HIV in the United States, and people on

cannot be shared publicly under order of Washinton Administration Code 246-101-635. Data is available upon request with data sharing agreement. Data requests can be made to the Washington State HIV Surveillance Program, which can be reached at hiv_surv@doh.wa.gov or (360) 236-3455.

**Funding:** JCD and CMK's work was supported by the University of Washington / Fred Hutch Center for AIDS Research, an NIH-funded program under award number AI027757 which is supported by the following NIH Institutes and Centers: NIAID, NCI, NIMH, NIDA, NICHD, NHLBI, NIA, NIGMS, NIDDK. There was no additional external funding received for this study The funders had no role in study design, data collection and analysis, decision to publish, or preparation of the manuscript.

**Competing interests:** I have read the journal's policy and the authors of this manuscript have the following competing interests: CMK has received donations of research supplies from Hologic, Inc. for activities outside of the submitted work

the program have high rates of viral suppression, the central measure of successful HIV treatment [2, 3]. Despite the benefits of the program, many PLWH struggle to complete the procedures required to maintain enrollment in the program and subsequently become disenrolled [4, 5]. However, the effect of ADAP disenrollment, particularly temporary disruptions in coverage, on HIV clinical outcomes is unclear.

In October of 2021, the Federal Ryan White HIV/AIDS program announced that they were removing the federal requirements for client recertification [6]. Prior to this, recipients of ADAP services needed to provide documentation of their eligibility every 6 months or they would be removed from the program and lose access to services [7]. This policy was identified as a barrier to client access to ADAPs. In a prior study, we found that 26% of ADAP enrollees in Washington State were disenrolled due to failure to recertify in a 2-year period [8]. If they so choose, state programs now have the authority to modify the recertification requirement which may improve retention in the program. However, retaining more clients in the program may increase the cost of the program, and the impact of disenrollment on HIV care has not been established.

There is evidence to suggest that disenrollment from ADAP may disrupt access to antiretroviral medications and HIV medical care. A 2018 HRSA request for information from Ryan White programs, clients, and providers identified the 6-month recertification requirement as a disruptor to HIV care [9]. Changes to insurance coverage have been demonstrated to be disruptive to ART use; a 2016 study found that PLWH who lose insurance were less likely to be virally suppressed [10, 11]. A loss of ADAP coverage may occur at a particularly vulnerable time for clients; if a person is unable to complete the recertification procedures, it may indicate that they have a reduced capacity to navigate medical and payer systems. To our knowledge there is no published data examining the effect of ADAP disenrollment on viral suppression or how this may be modified by a client's existing insurance coverage.

In this study we examined the change in viral suppression associated with ADAP disenrollment. The objectives of this study were to: 1) Compare the risk of viral suppression between those who disenroll in ADAP or are ruled ineligible for ADAP and those who are continuously enrolled; 2) Estimate the risk differences by co-insurance type; and 3) Describe the potential effect of confounding by mental health, binge-drinking, illicit substance use, and homelessness on the measured risk difference using quantitative bias analysis.

## Methods

### Study design and data sources

We performed a longitudinal analysis of PLWH enrolled in Washington State's ADAP between 2017 and 2019. To be eligible for inclusion in our analysis, ADAP clients had to either be continuously enrolled in ADAP until the end of 2019, receive a viral load in Washington state within 12 months of leaving ADAP, or live in Washington state for 12 months after leaving ADAP. Residency in Washington state was inferred from subsequent laboratory reports or interstate deduplication efforts based on Soundex that occur when a person has not received a lab in Washington for 18 months.

The primary data sources were Washington HIV surveillance system and the Washington Ryan White Data System. The Washington HIV surveillance system is a longitudinal database of PLWH based on laboratory results collected during the course of routine HIV care and reported to the state by law [12]. The laboratory results are supplemented by active collection of additional information about new HIV diagnoses, changes of residence, death, and demographics which yields a data system that can be used to describe the population of PLWH in Washington and follow individuals' clinical outcomes over time. The Ryan White data system

is an administrative database of eligibility, benefits, and services provided for all Ryan White-funded activities in Washington State that is used for claims administration, eligibility determination, enrollment, program administration, and program evaluation.

The exposure of interest was recertification success, which we measured using data from the Ryan White data system. We characterized each client's recertification success over the study period by categorizing them as 1) continuously enrolled (i.e. the client successfully recertified throughout the two year time period), 2) disenrolled (i.e. the client failed to recertify one or more times), or 3) ineligible (i.e. the client submitted recertification information but did not meet the requirements for the program one or more times). More information about how clients were categorized is described elsewhere [8]. If a client was both disenrolled and ruled ineligible during the study period, they were assigned the status corresponding to the event that occurred first.

Our outcome was viral suppression status before and after clients' recertification opportunities, which we measured using data from the HIV surveillance system. Viral suppression is the central indicator of successful treatment of HIV and represents a health state with a reduced risk of HIV-associated illness and no risk of HIV transmission [13]. It is assessed via viral load testing, which is generally performed every 6 to 12 months for PLWH in continuous care. We measured viral suppression before a recertification opportunity by the result of a client's last viral load test before the recertification date and measured viral suppression after a recertification opportunity by the result of the viral load test immediately following their recertification date. We used a cutoff of 200 viral copies per milliliter and considered clients to be not suppressed if they did not have a viral load test in the 12 months preceding or following a recertification [14].

To characterize the demographics of our population, we extracted age, race and ethnicity, sex at birth, HIV acquisition risk (male sex with male, injection drug use, male sex with male and injection drug use, heterosexual contact, and other), and region (Western Washington, Eastern Washington, or Seattle/King County) from the HIV surveillance system and insurance type (private, dual Medicare-Medicaid, other public insurance, uninsured) and enrollment in case management from the Ryan White data system. For insurance type, "other public insurance" primarily consists of Medicare plans, but also includes VA and Indian Health Service coverage. Individuals with Medicaid are generally not eligible for ADAP (apart from those with dual Medicare-Medicaid). We presented the frequencies of each characteristic of the population by recertification outcome.

## Primary analysis

We estimated the risk difference in viral suppression associated with disenrollment and ineligibility rulings using a generalized linear model with the Poisson distribution, an identity link function, and subject-level random effects [15, 16]. With time as a binary variable indicating before or after a recertification opportunity using the following model equation:

$$P(\text{Viral Suppression} = 1 | \text{Disenrolled, Ineligible, Time})$$
$$= \alpha + \beta_1 * \text{Disenrolled} + \beta_2 * \text{Ineligible} + \beta_3 * \text{Time} + \beta_4 * \text{Disenrolled} * \text{Time}$$
$$+ \beta_5 * \text{Ineligible} * \text{Time} + \beta_6 * \text{Covariates}$$

Each ADAP client contributed one recertification opportunity to the model; although individual clients may have encountered multiple recertification opportunities during our study period, we chose to analyze only a single event to ensure that we did not re-use individual viral load test results across observations. For clients categorized as disenrolled or ineligible, we modelled viral suppression surrounding the date they were first disenrolled or ruled ineligible.

For clients who were continuously enrolled, we modelled viral suppression surrounding their first successful recertification. We constructed adjusted models that included age (linear splines with cut points at 25, 50, and 75), sex, race/ethnicity, HIV transmission category, region, insurance type, and case management status. As the value of ADAP services may depend on a client's ability to afford care via other means, we repeated our analyses with an interaction term between the disenrollment and ineligibility terms and insurance type to estimate risk differences specific to insurance type. This was an a priori decision. We also performed a sensitivity analysis where we excluded clients who did not have a viral load before or after their recertification opportunity of interest. (In primary analyses we categorized individuals who did not have labs as not virally suppressed). This was motivated by concern about individuals moving out of Washington; laboratory results for someone who has moved away would not be reported to the Washington State Department of Health and can give the appearance that the person is not receiving medical care. This may be overrepresented among people who disenroll, as people who leave the state likely would not apply for ADAP recertification. To quantify the impact of this analytic decision, we repeated our analysis while excluding clients who did not have a viral load before or after their recertification opportunity of interest. We also described the median and quartiles of number of months between viral load testing and recertification opportunity and the percent of clients who had zero, one, and 2 or more viral load tests before and after their recertification opportunities.

## Quantitative bias analysis

Quantitative bias analysis is a technique for describing the influence of systematic error on an epidemiologic study's estimand of association [17]. In our study, we used quantitative bias analysis to examine the impact of unmeasured confounders. When individual-level confounder information is not available, quantitative bias analysis can be used to simulate adjusted estimates using descriptions of population characteristics from external data sources. Through review of the literature and consultation of HIV care providers, we identified four variables that we could not measure directly for the entire analytic sample, but may be confounders of the association between ADAP disenrollment and viral suppression: poor mental health [18], heavy drinking [19], illicit substance use [20], and housing instability [21]. To develop adjusted estimates three parameters are required: 1.) the risk difference of viral suppression between those who were exposed to each confounder and those who were not exposed to each confounder among those who were in ADAP, 2.) the prevalence of the confounders among those who remained enrolled in ADAP, and 3.) the prevalence of the confounders among those who disenrolled from ADAP (Fig 1) [17]. To estimate these parameters, we used data collected from current or former ADAP clients who participated in the Washington Medical Monitoring Project (MMP) between 2015 and 2018. MMP is a surveillance system that captures detailed information on behavioral and clinical characteristics of a sample of people living with HIV in the US via structured phone interview and medical record abstraction [22]. The data is collected by health department staff, who are assigned a random sample of PLWH to contact via letter and phone call. The data is accompanied by weights to correct for non-response bias. We measured our confounders using the following interview questions:

- Poor Mental health [18]- Yes/No, reflecting a participant's answer to the question, "Now thinking about your mental health, which includes stress, depression, and problems with emotions, for how many days during the past 30 days was your mental health not good?", using a cutoff of 14 or more days as poor mental health [23].

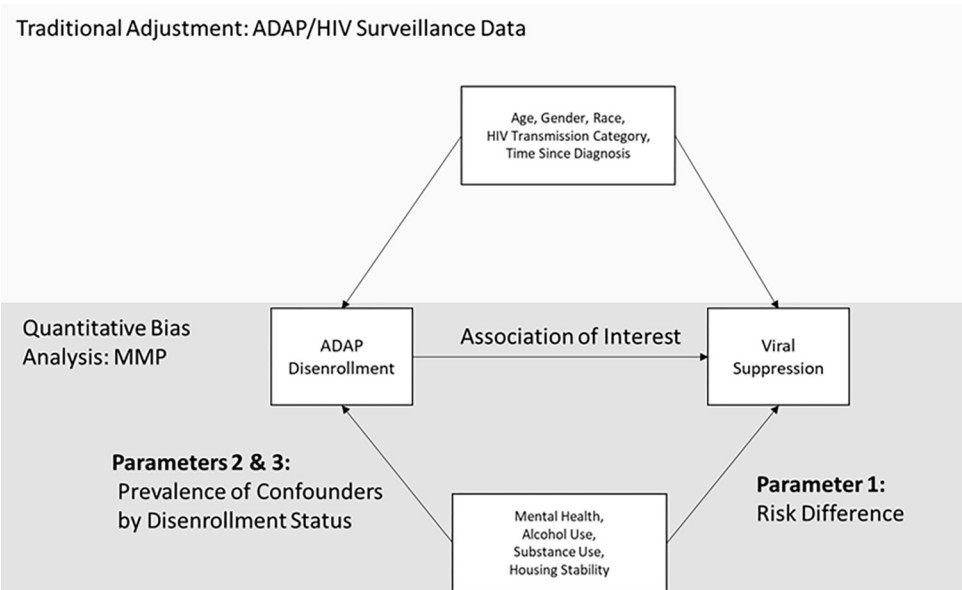

**Fig 1. Quantitative bias parameters to adjust the relationship between ADAP disenrollment and viral suppression for key barriers to engagement in HIV care.**

- Heavy Drinking [19]- Yes/No, reflecting the CDC criterion of "heavy drinking" in the past 30 days, which is an average of more than 2 alcoholic drinks per day for men, more than 1 alcoholic drink per day for women.

- Illicit Substance use [20]- Yes/No reflecting whether a participant reported using cocaine, heroin, or methamphetamine in the past 12 months

- Housing Instability [21]- Yes/No reflecting whether a participant reported living on the street, shelter or car; needing housing, rent, or utility assistance; or living with friends in the past 12 months.

To estimate the first parameter, we used a single Poisson model (including all 4 confounders) to estimate the risk difference of viral suppression at the time of interview for each confounder. We derived parameter 2 from the proportion of our MMP sample who selected 'Yes' for each confounder. For parameter 3, we simulated a range of values ranging from 0% to 100% higher than parameter 2. We used these 3 parameters together to calculate corrected risk differences using the methodology described in Lash et al. [17]. We obtained 95% confidence intervals using 10,000 Monte Carlo simulations. We used a normal distribution to represent uncertainty in our Parameter 1 & 2 estimates using the standard errors from our regression model and the asymptotic standard errors of the binomial distribution. We treated parameter 3 as a fixed value. We estimated a corrected risk difference using each confounder individually and for all confounders together.

All analyses using MMP data were performed with subject-specific weights to account for non-response. All analyses met the criteria for program evaluation and received exemption from review from the University of Washington IRB. Data was collected as surveillance activities as mandated by Washington state law; informed consent was not required or obtained. Data was anonymized during analysis, but not before access as the study team is responsible for data collection.

## Results

From 2017–2019, 5480 clients were enrolled in ADAP, of whom 5238 (96%) met our inclusion criteria; 3006 (55% of ADAP enrollees) remained in ADAP through the end of 2019 and 2231 (40% of ADAP enrollees) received a viral load or lived in WA for 12 months after leaving ADAP. Of the eligible population, 1336 (26%) were categorized as disenrolled, 896 (17%) as ineligible, and 3006 (57%) as continuously enrolled. The characteristics of this population are presented in Table 1.

In the period immediately preceding the recertification opportunity, 1109 (83%) of those who were disenrolled were virally suppressed, 806 (90%) of those who were ruled ineligible were virally suppressed, and 2756 (92%) of those who were continuously enrolled were virally suppressed. Following removal from the program, 927 (69%) of those who were disenrolled were virally suppressed (adjusted risk difference 12%, 95% CI 9–15%) and 730 (81%) of those who were ruled ineligible were virally suppressed (adjusted risk difference 7%, 95% CI 4–10%). Following recertification, 2700 (90%) of those who were continuously enrolled were virally suppressed (Table 2).

For those who were disenrolled, the adjusted risk difference of viral suppression was highest among clients with dual Medicare/Medicaid (22%, 95% CI 9–35%), and lowest among those with private insurance (8%, 95% CI 5–12%). For those who were ineligible, the highest risk difference was also among clients with other public insurance (21%, 95% CI 11–33%), but was lowest among the uninsured (4%, 95% CI -5-13%).

Our sensitivity analysis excluding participants who did not have a viral load measurement before or after their recertification yielded attenuated risk differences. Among those who received viral load tests 12 months before and after their recertification date, the adjusted risk difference of viral suppression was 3% (95% CI 1–5%) for those who were disenrolled and was 2% (95% CI 0–4%) for those who were ruled ineligible.

For those who received viral load testing within a year recertification, the median time to testing was 3 months and was the same for those who recertified, dropped, or were ruled ineligible. Full information about frequency of viral load testing can be found in Table 3.

### Quantitative bias analysis

Between 2015 and 2018, there were 308 MMP participants who were current or previous clients of ADAP. From a model including each variable, the risk difference for viral suppression was -13.8% (95% CI -1.94 to -8.1%) for unstable housing, -2.3% (95% CI -13.6 to -8.1%) for poor mental health, -4.1% (95% CI -24.3 to 16.1%) for heavy drinking, and 3.8% (95% CI -8.7 to 16.4%) for illicit substance use. The prevalence of these confounders was 39%, 23%, 5%, and 16%, respectively. If the prevalence of these confounders were 50% higher among those who disenroll from ADAP, the corrected risk difference for viral suppression from our main analysis would be 9% (95% CI 5–13%) instead of the 12% unadjusted risk difference. If the prevalence of these confounders were 100% higher among those who disenroll from ADAP, the corrected risk difference for viral suppression would be 6% (0–12%) (see Fig 2). The effect of the individual variables is shown in Table 4.

### Discussion

We found that 12 out of every 100 PLWH who were disenrolled from ADAP lost viral suppression due to their disenrollment. This effect was largest among clients who had dual Medicaid/Medicare insurance (22/100) and smallest among clients who had private insurance (8/100). Our quantitative bias analysis showed that the barriers to viral suppression of housing instability, poor mental health, binge-drinking, and illicit substance use partially explain but do not

**Table 1. Demographic characteristics of ADAP clients who were disenrolled, ruled ineligible, and were continuously enrolled, Washington state 2017–2019ᵃ.**

| Variable | Value | Disenrolled | Ruled Ineligible | Continuously Enrolled |
|---|---|---|---|---|
| Total | | 1336 | 896 | 3006 |
| Race/ethnicity | Hispanic | 267 (20%) | 175 (20%) | 637 (21%) |
| | Black | 297 (22%) | 205 (23%) | 489 (16%) |
| | White | 636 (48%) | 407 (45%) | 1562 (52%) |
| | Multiple | 136 (10%) | 109 (12%) | 318 (11%) |
| HIV Acquisition Risk Category | MSM | 807 (60%) | 543 (61%) | 1816 (60%) |
| | IDU | 61 (5%) | 31 (3%) | 124 (4%) |
| | MSM+IDU | 122 (9%) | 84 (9%) | 281 (9%) |
| | Heterosexual Contact | 122 (9%) | 90 (10%) | 333 (11%) |
| | Unknown | 224 (17%) | 148 (17%) | 452 (15%) |
| Age | <25 | 23 (2%) | 16 (2%) | 38 (1%) |
| | 25–34 | 212 (16%) | 173 (19%) | 251 (8%) |
| | 35–44 | 291 (22%) | 227 (25%) | 481 (16%) |
| | 45–54 | 371 (28%) | 259 (29%) | 774 (26%) |
| | 55–64 | 313 (23%) | 176 (20%) | 959 (32%) |
| | 65+ | 126 (9%) | 45 (5%) | 503 (17%) |
| Birth Sex | Female | 210 (16%) | 161 (18%) | 502 (17%) |
| | Male | 1126 (84%) | 735 (82%) | 2504 (83%) |
| Insurance | Dual Medicare-Medicaid | 66 (5%) | 24 (3%) | 316 (11%) |
| | Other Public | 761 (57%) | 668 (75%) | 1286 (43%) |
| | Private | 308 (23%) | 72 (8%) | 1170 (39%) |
| | Uninsured | 201 (15%) | 132 (15%) | 234 (8%) |
| Geography | Eastern WA | 229 (18%) | 135 (15%) | 714 (24%) |
| | King County | 748 (57%) | 508 (57%) | 1463 (49%) |
| | Western WA (Not King County) | 327 (25%) | 245 (28%) | 807 (27%) |
| Time from Diagnosis | <1 year | 139 (10%) | 105 (12%) | 351 (12%) |
| | 1–5 years | 240 (18%) | 183 (20%) | 336 (11%) |
| | >5 years | 957 (72%) | 608 (68%) | 2319 (77%) |
| Income | 0–135% FPL | 437 (33%) | 168 (19%) | 1179 (39%) |
| | 135–250% FPL | 446 (33%) | 288 (32%) | 1071 (36%) |
| | 250–425% FPL | 452 (34%) | 440 (49%) | 756 (25%) |

Abbreviations: MSM = Male-Male Sexual Contact, IDU = Injection Drug Use, FPL = Federal Poverty Level

a. Clients who were ruled ineligible one or more times between 2017 and 2019 were categorized as ineligible unless they were previously categorized as disenrolled. Clients who were never removed from ADAP were categorized as continuously enrolled. Time varying variables (age, insurance, geography, income, and time from diagnosis) were measured at a client's first disenrollment for those who disenrolled and first recertification opportunity for those who were continuously enrolled.

entirely account for the effect. This suggest that loss of ADAP coverage has a direct effect on reducing viral suppression.

The large drop in the proportion virally suppressed for those who are disenrolled from the program suggest that a subset of clients who are removed from ADAP are left without a way to access medications. This is consistent with ADAP's role as a 'last resort' payer and a survey of PLWH in Alabama, of whom 21% had lapses in treatment due to problems with ADAP or medication costs [24]. The larger effect sizes among individuals with dual Medicare/Medicaid (who necessarily have a disability) and uninsured individuals (who may have to pay full price for medical care) are consistent with a situation where a subset of the clients who are disenrolled do not have the resources to continue accessing HIV care. It is noteworthy that there

**Table 2. Viral suppression status before and after removal from ADAP or reenrollment by insurance status, Washington state 2017–2019.**

| Population | Recertific-ation Outcome[a] | N Total | Virally Suppressed Before Recertification or Removal[b] | Virally Suppressed After Recertification or Removal | Change in Viral Suppression | Unadjusted Risk Difference[c] | Adjusted Risk Difference[d] |
|---|---|---|---|---|---|---|---|
| **All** | Disenrolled | 1336 | 1109 (83%) | 927 (69%) | 14% (10–17%) | 12% (9–15%) | 12% (9–15%) |
| | Ineligible | 896 | 806 (90%) | 730 (81%) | 8% (5–12%) | 7% (4–10%) | 7% (4–10%) |
| | Enrolled | 3006 | 2756 (92%) | 2700 (90%) | 2% (0–3%) | Reference | Reference |
| **Uninsured** | Disenrolled | 201 | 142 (71%) | 118 (59%) | 12% (3–21%) | 9% (1–18%) | 12% (3–20%) |
| | Ineligible | 132 | 102 (77%) | 96 (73%) | 5% (-6-14%) | 2% (-7-11%) | 4% (-5-13%) |
| | Enrolled | 234 | 204 (87%) | 198 (85%) | 3% (-3-9%) | Reference | Reference |
| **Private** | Disenrolled | 763 | 668 (88%) | 594 (78%) | 10% (6–14%) | 9% (5–12%) | 8% (5–12%) |
| | Ineligible | 670 | 618 (92%) | 569 (85%) | 7% (4–11%) | 6% (3–9%) | 6% (3–10%) |
| | Enrolled | 1289 | 1205 (93%) | 2288 (92%) | 1% (0–3%) | Reference | Reference |
| **Dual Medicare/ Medicaid** | Dropped | 66 | 52 (79%) | 36 (55%) | 24% (9–40%) | 22% (9–35%) | 22% (9–35%) |
| | Ineligible | 24 | 21 (88%) | 17 (71%) | 17% (-6-39%) | 14% (-8-4%) | 16% (-8-39%) |
| | Enrolled | 316 | 277 (88%) | 270 (85%) | 2 (-3-8%) | Reference | Reference |
| **Other Public Insurance** | Dropped | 308 | 248 (81%) | 181 (59%) | 22% (15–29%) | 20% (13–26%) | 19% (14–26%) |
| | Ineligible | 72 | 67 (93%) | 50 (69%) | 24% (11–36%) | 21% (10–32%) | 22% (11–33%) |
| | Enrolled | 1170 | 1072 (92%) | 1046 (89%) | 2% (0–5%) | Reference | Reference |

a. Clients who were ruled ineligible one or more times between 2017 and 2019 were categorized as ineligible unless they were previously categorized as disenrolled. Clients who were never removed from ADAP were categorized as continuously enrolled.

b. Viral suppression measured after a client's first removal from ADAP or first recertification if they were never removed. Clients were categorized as virally suppressed before recertification if they had a viral load of 200 copies/mL or less in the 12 months prior to the recertification. Clients were categorized as virally suppressed after recertification if they had a viral load of 200 copies/mL or less in the 12 months after recertification.

c. Risk difference from a generalized linear model with the Poisson distribution, an identity link function, and subject-level random effects.

d. Adjusted risk difference from model adjusted for age, region, HIV acquisition risk, race/ethnicity, sex at birth, receipt of case management services, and insurance type (for non-stratified models).

was also significant decrease in viral suppression among those who were ruled ineligible from ADAP. This population is presumed to have alternative mechanisms to pay for healthcare (an income>425% FPL or Medicaid), but these results suggest that disruption of or change to medical coverage may still adversely impact this population.

Our quantitative bias analysis demonstrates that the major barriers to care that we measured (mental health, substance use, and housing instability) are not sufficient to explain the associations we observed, unless the prevalence of these factors are more than twice as high among those who are disenrolled as those who are not. This would require an extremely high prevalence of certain comorbidities among those who disenroll (e.g. over 78% homelessness

**Table 3. Frequency of viral load testing by ADAP enrollment status, Washington state 2017–2019.**

| Number of Viral Load Tests in 12 Month Period | Before Recertification | | | After Recertification | | |
|---|---|---|---|---|---|---|
| | **Disenrolled** | **Ineligible** | **Enrolled** | **Disenrolled** | **Ineligible** | **Enrolled** |
| 0 | 106 (9%) | 42 (5%) | 123 (4%) | 274 (21%) | 95 (11%) | 172 (6%) |
| 1 | 357 (31%) | 228 (29%) | 568 (20%) | 370 (28%) | 281 (31%) | 673 (22%) |
| 2+ | 692 (60%) | 520 (66%) | 2161 (76%) | 692 (52%) | 520 (58%) | 2161 (72%) |
| Time Between Recertification and Viral Load Testing | | | | | | |
| Median (Q1, Q3) | 4 (2,6) | 4 (2,6) | 3 (2,5) | 3 (2,6) | 3 (2,6) | 3 (2,5) |

a. Clients who were ruled ineligible one or more times between 2017 and 2019 were categorized as ineligible unless they were previously categorized as disenrolled. Clients who were never removed from ADAP were categorized as continuously enrolled.

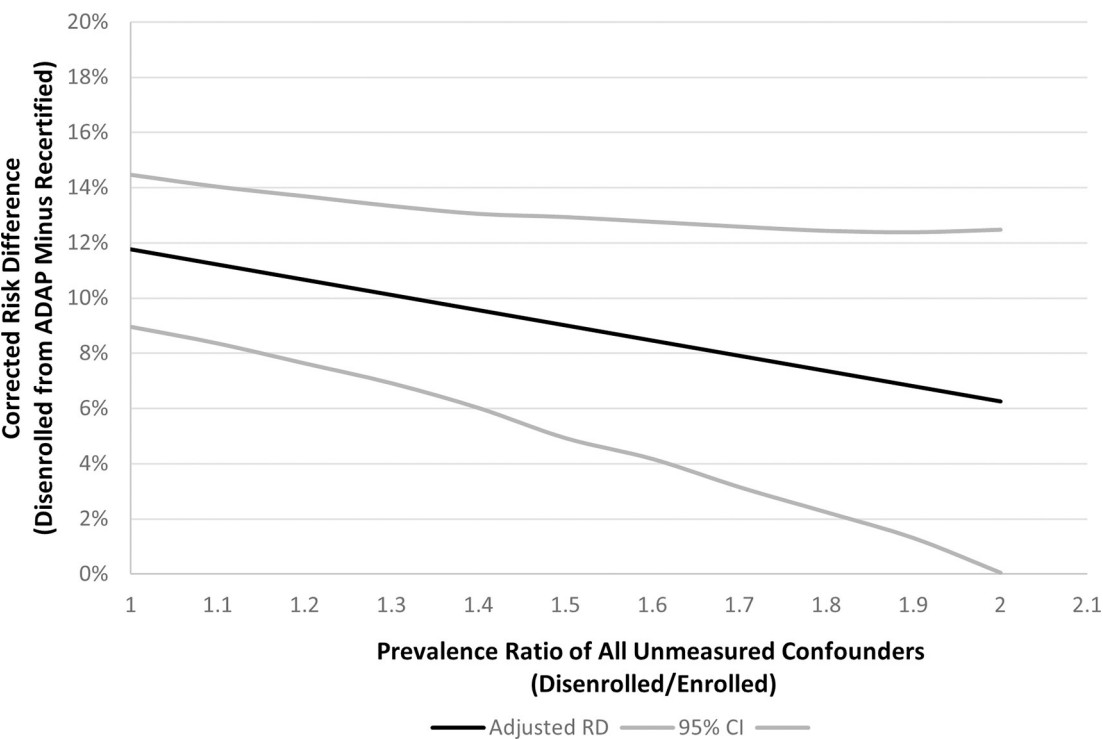

**Fig 2. Risk Difference (RD) of viral suppression from ADAP disenrollment and viral suppression after correction for unstable housing, poor mental health, heavy drinking, and illicit substance use, Washington state 2017–2019.**

and 32% illicit substance use in the past 12 months) which seems unlikely. Our findings provide evidence that disenrollment is a direct cause of clients falling out of HIV care, rather than simply a consequence of the same barriers that cause PLWH to fall out of HIV care or stop antiretroviral medications. This is consistent with responses to the HRSA request for information about the Ryan White program's administrative burdens, where program staff and HIV care providers described the recertification requirement as a barrier to keeping clients in care, and with literature that found insurance changes to be disruptive to HIV care more generally [9, 11].

Our study had several limitations. The first is related to our measurement of viral suppression. Some people who have left Washington state or who only require infrequent viral load monitoring as may be misclassified as virally unsuppressed. However, our sensitivity analysis that includes only people who have labs reported in Washington confirms the presence of a risk difference even in a situation where this misclassification is not possible. The variables used in our quantitative bias analysis are imperfect measures and come from a data source (MMP) that may rely on a biased sample. As such there likely is residual confounding or additional confounders that bias our estimates. However, it would require a very strong confounder or many additional weak confounders to explain the entire remaining effect after controlling for variables in our analysis. Finally, it is possible that some clients who do not recertify choose not to do so because they know that they are ineligible. However, this would only serve to attenuate the risk difference among those disenrolled, as it would cause clients in our study who are ineligible to be misclassified as disenrolled.

The generalizability of these findings to other ADAP programs depend on the similarity of their ADAP benefits and the care alternatives for people who leave ADAP. Washington's ADAP offers a wider range of services than many other state ADAP programs, some of which

**Table 4. Probabilistic quantitative bias analysis for unmeasured confounders of ADAP disenrollment and viral suppression, Washington state 2017–2019[a].**

| Confounder[b] | Prevalence Among MMP Participants in ADAP (SE) | Estimated Risk Difference (SE) of Viral Suppression[c,d] | Corrected Risk Difference of Viral Suppression with Assumption of Prevalence Among Disenrolled vs Continuously Enrolled | | |
|---|---|---|---|---|---|
| | | | 0% Greater Prevalence Among APAP Disenrollees | 50% Greater Prevalence Among APAP Disenrollees | 100% Greater Prevalence Among APAP Disenrollees |
| Unstable Housing | 39% (0.31) | 13.8% (0.053) | 12% (9–15%) | 9% (6–12%) | 6% (1–11%) |
| Poor Mental Health | 23% (0.26) | 2.3% (0.058) | 12% (9–15%) | 11% (8–15%) | 11% (7–15%) |
| Heavy Drinking | 5% (0.14) | 4.1% (0.103) | 12% (9–15%) | 11% (9–15%) | 11% (9–15%) |
| Illicit Substance Use | 16% (0.23) | -3.8% (0.064) | 12% (9–15%) | 15% (11–19%) | 18% (12–24%) |
| All | - | - | 12% (9–15%) | 9% (5–13%) | 6% (0–12%) |

a. Quantitative bias analysis performed using the prevalence of the unmeasured confounders among ADAP participants, the risk difference for viral suppression between those with and without the unmeasured confounders, and assigned values for prevalence of the unmeasured confounders among those who are disenrolled from ADAP. 95% confidence intervals were calculated using a normal distribution for the parameters and Monte Carlo sampling.

b. Unstable housing defined as living on the street, shelter or car; needing housing, rent, or utility assistance; or living with friends in the past 12 months. Poor mental health defined as having 14 or more self-reported days or poor mental health in the past 30 days. Heavy drinking defined as consuming more than 2 alcoholic drinks per day for men or more than 1 alcoholic drink per day for women on average in the past 30 days. Illicit substance use defined as use of cocaine, heroin, or methamphetamines in the past 12 months.

c. Viral suppression measured after a client's first removal from ADAP or first recertification if they were never removed. Clients were categorized as virally suppressed before recertification if they had a viral load of 200 copies/mL or less in the 12 months prior to the recertification. Clients were categorized as virally suppressed after recertification if they had a viral load of 200 copies/mL or less in the 12 months after recertification.

d. Risk difference from a generalized linear model with the Poisson distribution and identity link function that included all 4 confounders.

can pay for little more than direct HIV care [25]. Although the range of services in Washington undoubtedly contribute to clients' ability to achieve viral suppression, it also means that some clients in Washington ADAP may only enroll in ADAP for supplemental services. This suggests that disenrollment from ADAP in other states could have a larger impact on viral suppression, as in some cases every client would experience changes to their access to ART, rather than only the portion of clients in Washington who use HIV prescription benefits. Further, Washington is a Medicaid expansion state, which means that the lowest income populations in the state are in Medicaid rather than ADAP. In non-expansion states, where it is more difficult to enroll in Medicaid, ADAP may be serving populations that have fewer resources to accommodate disenrollment. Taken together, this suggests that the impact of disenrollment on viral suppression may be the same or larger in other jurisdictions.

We found that in Washington State, a significant proportion of PLWH who were removed from ADAP due to failure to recertify lost viral suppression immediately afterwards. This suggests that the default 6-month recertification policy serves as a barrier to achieving federal goals for HIV incidence and to the health of PLWH generally. In light of these findings, we recommend that ADAP programs make use of the new flexibility in federal recertification policy and examine alternatives to the 6-month recertification requirement that improve retention in their programs.

## Author Contributions

**Conceptualization:** Steven J. Erly, Christine M. Khosropour, Julia C. Dombrowski.

**Data curation:** Steven J. Erly, Jennifer R. Reuer.

**Formal analysis:** Steven J. Erly.

**Investigation:** Steven J. Erly, Julia C. Dombrowski.

**Methodology:** Steven J. Erly, Christine M. Khosropour, Anjum Hajat, Monisha Sharma, Julia C. Dombrowski.

**Supervision:** Christine M. Khosropour, Anjum Hajat, Monisha Sharma, Jennifer R. Reuer, Julia C. Dombrowski.

**Writing – original draft:** Steven J. Erly.

**Writing – review & editing:** Christine M. Khosropour, Anjum Hajat, Monisha Sharma, Jennifer R. Reuer, Julia C. Dombrowski.

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
