## [Decision Letter · Decision Letter 0]

13 Jan 2023

PONE-D-22-13718AIDS Drug Assistance Program Disenrollment is Associated with Loss of Viral Suppression beyond Differences in Homelessness, Mental Health, and Substance Use Disorders: An Evaluation in Washington State 2017-2019PLOS ONE

Dear Dr. Erly,

Thank you for submitting your manuscript to PLOS ONE. After careful consideration, we feel that it has merit but does not fully meet PLOS ONE’s publication criteria as it currently stands. Therefore, we invite you to submit a revised version of the manuscript that addresses the points raised during the review process. The manuscript has been evaluated by two reviewers, and their comments are available below.

The reviewers have raised concerns regarding the reporting and methodology of this study. 

Could you please revise the manuscript to carefully address the concerns raised?

We look forward to receiving your revised manuscript.

Kind regards,

Johannes Stortz

Staff Editor

PLOS ONE

Journal Requirements:

2. In your ethics statement in the manuscript and in the online submission form, please provide additional information about the patient records used in your retrospective study. Specifically, please ensure that you have discussed whether all data were fully anonymized before you accessed them.

“JCD and CMK's work was supported by the University of Washington / Fred Hutch Center for AIDS Research, an NIH-funded program under award number AI027757 which is supported by the following NIH Institutes and Centers: NIAID, NCI, NIMH, NIDA, NICHD, NHLBI, NIA, NIGMS, NIDDK. https://depts.washington.edu/cfar/?q=home”

Reviewers' comments:

Reviewer's Responses to Questions

**Comments to the Author**

1. Is the manuscript technically sound, and do the data support the conclusions?

Reviewer #1: Yes

Reviewer #2: Yes

2. Has the statistical analysis been performed appropriately and rigorously? 

Reviewer #1: Yes

Reviewer #2: Yes

3. Have the authors made all data underlying the findings in their manuscript fully available?

Reviewer #1: Yes

Reviewer #2: No

4. Is the manuscript presented in an intelligible fashion and written in standard English?

Reviewer #1: Yes

Reviewer #2: Yes

5. Review Comments to the Author

Reviewer #1: Thank you for the opportunity to review. The authors provide a careful analysis of changes in HIV viral suppression after disenrollment from Washington State’s AIDS Drug Assistance Program. The findings are directly relevant to policy, following the October 2021 removal of federal requirements that clients recertify every 6 months. The writing is clear and the quantitative bias analysis using the MMP data enhances the results. I have a few suggestions for the authors as they revise their work.

First, please add a “Table 1” of descriptives of the main study sample. Some of this information is in Supplementary Table 1. I recommend moving that to the main text and adding a column to provide descriptives on the full sample. In the Supplementary Table 1, the prevalence ratio is less intuitive for illustrating the differences between the two groups. As your reader, I am unsure how to interpret this column to quickly understand how characteristics differ between groups. Is there an easier way to present p-values showing differences between groups? If you prefer this format, please explain in the footnote and text how to interpret.

Second, I recommend a flow chart or other visualization to explain the sample design. Starting on line 136, the authors explain that each client contributed one recertification opportunity. If recertification were required every 6 months during the study period (per federal rules in place during the 2017-2019 study period), a client could potentially contribute multiple observations. It would be useful to have a better understanding of the time period of the study (are most observations from 2017 in the initial event?) and explanation for only considering one observation per study participant.

Third, a challenge with the HIV VL data is that in practice, VL testing is not always done precisely every 6 months. The VL tests can be missing for multiple reasons: falling out of care (which the authors code as not virally suppressed), having the tests done out of state and not reported to Washington’s surveillance system (for example, for persons who spend winter months in a different state), or because the patient is stable on their medications thereby receiving less frequent VL testing than the clinical guidelines (a comment I have heard anecdotally from clinicians). The authors already have a sensitivity analysis that excludes participants without a VL measurement before or after recertification. It would be useful to have more details on the timing of VL measurements in the data such as the percent of participants with 1+ or 2+ VL tests per year, the average and range in the time in between disenrollment & recertification events and VL testing, and comparison of sub-populations with and without regular VL testing. Please also discuss in the limitations how the timing of VL tests in relation to the time of recertification & disenrollment events might bias findings.

Reviewer #2: In this manuscript, the authors link data from the Washington State ADAP program with Washington State HIV surveillance data to understand the impact of disenrollment on viral suppression among ADAP enrollees between 2017 to 2019. They found that disenrollment was associated with a 12% decrease in viral suppression compared with continuous ADAP enrollment. They also found differences in the change in viral suppression by insurance status, with dual Medicare-Medicaid beneficiaries and those with “other public insurance” the most affected by disenrollment. Based on their findings, they conclude that disenrollment negatively impacts viral suppression and that simplifying or eliminating ADAP recertification may help to improve viral suppression among ADAP enrollees.

This is a well-written paper using unique datasets to answer an important policy question: what is the effect of ADAP disenrollment on viral suppression in Washington State? This question has not been addressed in the literature to date, making this both a unique and timely paper given the removal of federal requirements for ADAP client recertification in October 2021.

Major comments:

1. I appreciate the use of the risk difference to quantify the attributable risk of viral suppression due to lack of recertification. This is not used often enough for population-based studies such as this one.

2. Methods, lines 84-87: The inclusion criteria as described for this study is a little confusing. How is it known whether a client lives in Washington State for 12 months after leaving ADAP if they drop out of care? Is there active follow-up of those who drop out? Since this analysis uses surveillance data, it would seem that the residency status (Washington residence versus another state) of someone who disenrolls and does not have any contact with the providers in Washington State for over a year would be difficult to ascertain.

3. In addition to the important message about disenrollment and viral suppression, there may be a message about ineligibility and viral suppression worth mentioning in the Discussion, since those who became ineligible for ADAP also had substantial decreases in suppression (albeit in a smaller subset). However, some additional context may be necessary to better understand this (see my minor comment #1 below).

4. The comment in the Discussion that Washington is a Medicaid expansion state is an important one given that the inferences identified here could be exacerbated in a non-Medicaid expansion state. In fact, based on Table 1, “other public insurance” is the second largest group after “private insurance”. Can the authors clarify what type of insurance this is? Is it mostly Medicaid beneficiaries?

Minor comments:

1. Is there information on what the reasons for disenrollment and/or ineligibility among ADAP enrollees in Washington State, and how they are distributed? E.g., how often do ADAP clients lose eligibility because they obtain health insurance through a new job? Some context may be helpful to include in the Discussion if this information is available (although it may not be available).

2. Introduction, first sentence: The authors state that ADAPs are “a critical part of federal plans to reduce HIV incidence by 90% by 2030”. One assumes that this is because of the concept of “treatment as prevention” – this could be made clearer.

3. Introduction, line 60: The authors note that “individual programs now have the authority to modify the recertification requirement”. It would be helpful to insert the word “state” before the word “programs” to clarify and emphasize that ADAPs are administered by states.

4. Methods, line 107: if multiple VLs are recorded for a client before and after recertification opportunities, which ones are used in the analysis?

5. Methods, line 108: the word “is” is missing after “Viral suppression”

6. Methods, line 109: the word “of” is missing after “reduced risk”

7. Methods, line 156: consider replacing the word “estimate” with “estimand”

8. Methods, lines 171-172. Please clarify whether national or Washington State MMP data were used to estimate the bias parameters.

6. PLOS authors have the option to publish the peer review history of their article (what does this mean?). If published, this will include your full peer review and any attached files.

Reviewer #1: No

Reviewer #2: No

---

## [Author Response · Author response to Decision Letter 0]

24 Feb 2023

Please see the response to reviewers document for documentation of our revisions.

Thank you!

---

## [Decision Letter · Decision Letter 1]

6 Apr 2023

PONE-D-22-13718R1AIDS Drug Assistance Program Disenrollment is Associated with Loss of Viral Suppression beyond Differences in Homelessness, Mental Health, and Substance Use Disorders: An Evaluation in Washington State 2017-2019PLOS ONE

Dear Dr. Erly,

Thank you for submitting your manuscript to PLOS ONE. After careful consideration, we feel that it has merit but does not fully meet PLOS ONE’s publication criteria as it currently stands. Therefore, we invite you to submit a revised version of the manuscript that addresses the points raised during the review process.

The manuscript has undergone a thorough evaluation by two reviewers, and their comments are provided below. While the reviewer’s feel that most of their comments from the previous revision has been addressed there were some mathematical inconsistencies in the numbers reported within the tables of the Results section. We recommend thoroughly checking the numerical values for a accuracy.

We look forward to receiving your revised manuscript.

Kind regards,

Lucinda Shen, MSc

Staff Editor

PLOS ONE

Journal Requirements:

Reviewers' comments:

Reviewer's Responses to Questions

**Comments to the Author**

1. If the authors have adequately addressed your comments raised in a previous round of review and you feel that this manuscript is now acceptable for publication, you may indicate that here to bypass the “Comments to the Author” section, enter your conflict of interest statement in the “Confidential to Editor” section, and submit your "Accept" recommendation.

Reviewer #1: All comments have been addressed

Reviewer #2: All comments have been addressed

2. Is the manuscript technically sound, and do the data support the conclusions?

Reviewer #1: Yes

Reviewer #2: Yes

3. Has the statistical analysis been performed appropriately and rigorously? 

Reviewer #1: Yes

Reviewer #2: Yes

4. Have the authors made all data underlying the findings in their manuscript fully available?

Reviewer #1: No

Reviewer #2: No

5. Is the manuscript presented in an intelligible fashion and written in standard English?

Reviewer #1: Yes

Reviewer #2: Yes

6. Review Comments to the Author

Reviewer #1: The authors addressed all reviewer comments. The manuscript is clear, technically sound, and well executed. The authors do not make their data publicly available but it is surveillance data and likely cannot be shared publicly.

Reviewer #2: I appreciate the efforts that the authors made to address my comments. I note a few minor issues in the revised version that should be fixed.

1. Results section: The overall eligible N of 5237 does not match the sum of the component categories (1336 disenrolled, 896 ineligible, 3006 continuously enrolled). It is off by 1. Should the N be 5238?

2. Table 1: The “race” category should be changed to “race/ethnicity” since Hispanic is an ethnicity, not a race.

7. PLOS authors have the option to publish the peer review history of their article (what does this mean?). If published, this will include your full peer review and any attached files.

Reviewer #1: No

Reviewer #2: No

---

## [Author Response · Author response to Decision Letter 1]

10 Apr 2023

Thank you for taking the time to review our manuscript and giving us an opportunity to improve our work. We have revised the population size and the way we described race (now race/ethnicity) as the reviewers requested. More details are included in the "Response to Reviewers" document. Please let me know if there is anything else we can modify to improve our manuscript.

---

## [Editor Report · Decision Letter 2]

20 Apr 2023

AIDS Drug Assistance Program Disenrollment is Associated with Loss of Viral Suppression beyond Differences in Homelessness, Mental Health, and Substance Use Disorders: An Evaluation in Washington State 2017-2019

PONE-D-22-13718R2

Dear Dr. Erly,

We’re pleased to inform you that your manuscript has been judged scientifically suitable for publication and will be formally accepted for publication once it meets all outstanding technical requirements.

Kind regards,

Dario Ummarino, PhD

Senior Editor

PLOS ONE
---

## [Editor Report · Acceptance letter]

26 Apr 2023

PONE-D-22-13718R2 

AIDS Drug Assistance Program Disenrollment is Associated with Loss of Viral Suppression beyond Differences in Homelessness, Mental Health, and Substance Use Disorders: An Evaluation in Washington State 2017-2019 

Dear Dr. Erly:

I'm pleased to inform you that your manuscript has been deemed suitable for publication in PLOS ONE. Congratulations! Your manuscript is now with our production department. 

Kind regards, 

on behalf of

Miss Lucinda Shen 

Staff Editor

PLOS ONE